# Institutional investors' site visits and investment-cash flow sensitivity: Mitigating financing constraints or inhibiting agent conflicts?

Jia Liao[1], Yun Zhan[2]*, Yu Yuan[3]

1 Business School, Huaqiao University, Quanzhou, Fujian, China, 2 School of Economics, Jinan University, Guangzhou, Guangdong, China, 3 School of Management, Jinan University, Guangzhou, Guangdong, China

* 953185016@qq.com

**Data Availability Statement:** All relevant data are within the manuscript and its Supporting Information files.

## Abstract

Taking Chinese non-financial A-share companies listed on the Shenzhen Stock Exchange (SZSE) between 2003 and 2018 as a sample, this paper empirically examines whether and how institutional investors' site visits (SVs) affect corporate investment-cash flow sensitivity (ICFS). The results show that institutional investors' SVs can reduce ICFS, and this effect is more obvious for companies with fewer investment opportunities, larger sizes, higher internal cash flows, and higher agency costs, indicating that institutional investors' SVs primarily inhibit ICFS caused by agency conflicts rather than financing constraints. In addition, the inhibitory effect of institutional investors' SVs on ICFS exists mainly in companies with poor internal supervision governance and weak executive compensation incentive mechanisms, indicating that institutional investors' SVs and other forms of corporate governance mechanisms operate as substitutes in reducing ICFS. This paper reveals the important role of institutional investors' SVs in reducing ICFS, with important theoretical and practical implications for regulators to progressively regulate and promote this form of investor activity.

## 1. Introduction

As China's capital market gradually moves from brutal growth to high-quality development, regulatory authorities are increasingly committed to strengthening investor relations management of listed companies to provide favorable conditions for alleviating information asymmetry in the capital market and facilitating communication between investors and listed companies. In 2012, the Shenzhen Stock Exchange (SZSE) launched a disclosure website (Hudongyi website) and issued regulations that requiring companies to disclose the details of site visits (SVs) within two trading days of completion. Since then, corporate SVs have attracted widespread attention. Corporate SVs involve investors visiting a company to observe its production and operational processes, and meeting face-to-face with managers and employees, thus making it possible to obtain more useful and critical information about

**Funding:** The authors acknowledge the funding support by Fujian Provincial Social Science Foundation Project [grant number, FJ2024MGCA018]. The funders had no role in study design, data collection and analysis, decision to publish, or preparation of the manuscript.

corporate performance, corporate governance, and the sustainable development of the company [1, 2]. Existing research shows that SVs not only help investors gather information about companies and make informed transactions [3, 4], but also help managers learn from investors [5]. Thus, corporate SVs facilitate information exchange between management and investors [6], which helps curb opportunistic behaviors among management [2, 7, 8] and improve information efficiency in the capital market [9, 10].

Although all investors can conduct SVs on listed companies, retail investors hardly ever visit listed companies because the time and expense involved are uneconomical for them [11]. As important participants in the capital market, institutional investors provide a huge financial support to bolster the real economy [12], and contribute to promoting diversified information exchange in the capital market [13]. To make better investment decisions with a deeper understanding of the company, institutional investors' SVs and the economic consequences of their SVs have gradually become a hot issue in corporate governance research [14]. However, there is little literature on whether institutional investors' SVs affect corporate investment and financing behaviors. Therefore, this paper aims to take corporate investment-cash flow sensitivity (ICFS) as an entry point to investigate whether institutional investors can effectively play the role of external governance mechanisms and reduce ICFS to expand theoretical research in this area, and to provide important practical implications for regulators and investors.

Using the data from Chinese A-share companies listed on the SZSE between 2013 and 2018, this paper finds that capital investment is highly sensitive to companies' internal cash flows, while institutional investors' SVs can effectively reduce ICFS. This effect is more significant in the subsample with poor investment opportunities, large sizes, high internal cash flows, and high agency costs, indicating that institutional investors' SVs mainly inhibit ICFS caused by agency conflicts rather than by financing constraints. Further analysis finds that the above negative effect is concentrated in companies with poor internal supervision governance and weak executive compensation incentive mechanisms, revealing a substitution relationship between institutional investors' SVs and other forms of corporate governance mechanisms.

This paper makes two contributions. First, ICFS has been a key research topic in the field of corporate governance [15–18]. High ICFS not only restricts the sustainable development of enterprises but also leads to low efficiency of capital allocation in the whole society [19, 20]. Although studies have demonstrated that institutional investors can reduce ICFS [21, 22], no literature has focused on whether and how this particular behavior of institutional investors' SVs affects ICFS. This paper provides a thoughtful theoretical analysis and empirical test of the effects of institutional investors' SVs on ICFS in terms of both financing constraints and agency conflicts, extending the existing research on institutional investors and ICFS. Second, the economic consequences of institutional investors' SVs have become a hot topic in recent years [23], and existing studies have examined the impact of institutional investors' SVs on corporate innovation [11], earnings management [10], stock price crash risk [24], equity capital costs [25], dividend payouts [8], cash holdings [6], and corporate social responsibility [1]. This study confirms that substitution effects exist between institutional investors' SVs and other mechanisms of corporate governance in reducing ICFS, which expands the relevant research on institutional investors' SVs.

## 2. Literature review

Fazzari et al. [15] propose the "financing constraint hypothesis", arguing that the high ICFS is mainly due to the financing constraint caused by information asymmetry in capital markets, which is confirmed by numerous studies [17, 26–32]. Consistent with the idea that poorly governed companies have difficulty accessing external finance and therefore rely more on internal

funding, Francis et al. [33] find that ICFS increases with poor corporate governance. Using data on manufacturing companies from 1970 to 2006 sourced from Compustat, Brown and Petersen [34] find that ICFS decreases significantly as the stock market evolves. Drawing on 2,858 observations from the Vietnam Stock Market from 2009 to 2014, Thoa and Uyen [35] find that ICFS decreases after banking system reforms and that non-state-owned enterprises' underinvestment is mitigated by better accessibility to bank loans, while state-owned enterprises' overinvestment does not decrease. Guizani [36] applies the data of 84 Saudi-listed non-financial companies and observes that tight monetary policies, adverse financial developments, and liquidity crises exacerbate ICFS. Using firm-level data for 69 countries from 1995 to 2019, Wang [37] reports that companies in more liberalized financial markets exhibit lower ICFS and that alleviating financing constraints and then expanding the financing channels are potential mechanisms through which financial liberalization affects ICFS. However, Chen and Chen [16] and Machokoto et al. [18] both examine the time-series variation of ICFS and find that it tends to decline over time, indicating that the use of ICFS as a proxy for financial constraints is declining. Using a quasi-natural experiment with China's 4 trillion yuan stimulus package, Deng et al. [38] note a positive and significant relationship between ICFS and investment after controlling for financial constraints, confirming that ICFS cannot measure financing constraints.

Andren and Jankensgard [39] observe that ICFS decreases throughout abundance (2005–2008) for financially constrained companies, while it increases over time for financially unconstrained companies, revealing that the above relationship is driven by agency conflicts associated with internal cash flow. Using an unbalanced panel of Dutch companies, Degryse and de Jong [40] report that companies with lower Tobin's Q (i.e., facing the managerial discretion problem) have higher ICFS than companies with higher Tobin's Q (i.e., facing the asymmetric information problem), and, they also find that in the lower Tobin's Q subsample, ICFS is lower for companies with higher access to bank loans. Using data from Chinese listed companies from 2002 to 2005, Huang et al. [41] report that top executives' overconfidence increases ICFS, but this relation is observed only in companies with high agency costs. Kuo and Hung [42] find that ICFS is higher for family-owned companies with excess control rights due to the dominance of Type II agency conflicts, disentangling the effects of asymmetric information and agency conflicts caused by internal cash flow. Han and Pan [43] empirically test the impact of CEO internal debt on ICFS by analyzing a sample of US manufacturing companies from 2006 to 2012 and find that companies with higher CEO leverage ratios are significantly associated with higher ICFS. Drawing on 6,797 observations of listed companies in France from 2000 to 2013, Derouiche et al. [44] find that ICFS decreases with cash-flow rights but increases with the control rights of controlling shareholders. Peruzzi [45] investigates whether family ownership structure affects ICFS using Italian SMEs and finds that family companies have higher ICFS, and this relation is driven by the agency conflicts associated with ownership concentration and family management. Using a sample of Brazilian listed companies, Pellicani et al. [46] find that the family ownership structure does not directly affect ICFS of constrained companies while the active intervention of the controlling family on the board may aggravate agency conflicts and thus increase ICFS for unconstrained companies.

As the major actors in capital markets, institutional investors play significant roles in alleviating information asymmetry and inhibiting institutional conflicts [2, 11, 47]. Agca and Mozumdar [21] demonstrate a significant negative association between institutional ownership and ICFS by analyzing the data of U.S. manufacturing companies from 1970 to 2001, suggesting that institutional investors play a pivotal role in compensating for capital market deficiencies. Based on US companies, Attig et al. [22] conclude that institutional investors' investment horizon is significantly negatively correlated with ICFS because institutional

investors with longer investment horizons have stronger incentives to monitor effectively, and in turn, alleviates information asymmetry and agency conflicts. In recent years, institutional investors have devoted themselves to conducting SVs to deepen their understanding of listed companies and thus make better investment decisions, and the economic consequences of such behavior have gradually become a hot issue in corporate governance research [14]. From A-share listed companies on the SZSE, Jiang and Yuan [11] find that institutional investors' SVs significantly promote corporate innovation, and this positive relationship is more pronounced in companies with weak corporate governance and poor information environments, providing evidence that the information access behaviors of institutional investors can be complementary to other corporate governance mechanisms. Saci and Jasimuddin [25] use the unique datasets from Chinese listed companies on the SZSE from 2013 to 2017 and find that institutional investors' SVs can help companies achieve lower equity capital costs. Using 13,867 observations from A-share listed companies on the SZSE, Chen et al. [1] find that institutional investors' SVs can encourage visited companies to fulfill their social responsibility, and this effect is more pronounced in environments with weak legislative enforcements and religious atmospheres. Qi et al. [10] consider the detailed features of investors' SVs and find that accrual-based earnings management is negatively correlated with the number of external participants, particularly institutional investors, and the breadth and depth of communication between the two parties during visits. Using the data of companies listed on the SZSE from 2013 to 2019, Yang and Ma [8] observe that institutional investors' SVs significantly dampen dividend underpayment in firms with more serious agency conflicts or weaker corporate governance. Based on the datasets from A-share listed companies on the SZSE from 2012 to 2019, Wang et al. [6] find that institutional investors' SVs significantly increase corporate cash holdings and cash holding value.

## 3. Theory and hypothesis

Financing constraints severely discourage investment by companies with growth opportunities and induce underinvestment [15], while agency conflicts exacerbate negative NPV project investments by companies with excessive free cash flow and cause overinvestment [48]. All of the above will result in positive ICFS, which in turn will distort resource allocation efficiency [19, 32, 40, 49–51]. Over the past few years, there has been a marked increase in academic research dedicated to exploring the role of institutional investors' SVs in corporate governance [1, 8, 10, 11]. By sorting and summarizing the above two branches of literature, we argue that institutional investors' SVs influence ICFS through the following two channels.

Institutional investors' SVs can alleviate corporate financing constraints and thus inhibit ICFS. On the one hand, institutional investors' SVs can alleviate information asymmetry and reduce the financing difficulty and financing transaction costs of visited companies [25, 52]. Through conducting SVs, institutional investors go deep into the company, and directly observe the company's operating environment to fully understand the company's real business situation, but also with the management, employees, and other face-to-face communication and exchange, to maximize access to private information that is not publicly available [23]. The timely disclosure of information from institutional investors' SVs enables external investors to easily obtain information about a company's characteristics, which helps to reduce the asymmetry of information between internal and external companies [21]. An increase in information transparency between investors and firms will increase the efficiency of the capital market in interpreting firm information so that external investors can more accurately understand and grasp the operation and future development

of the firm's investment projects [53], thus inhibiting ICFS. On the other hand, institutional investors SVs can play an effective role in information mining and signaling, which helps attract more financial support for visited companies [54]. After conducting SVs, the entry and holding increase behaviors of institutional investors can send positive signals to the market, which in turn enhances the confidence and loyalty of existing or potential investors [55]. At this time, companies with good investment opportunities will be more likely to attract the attention of investors, and external investors will be more optimistic in their assessment of the investment project, which can help the company obtain external financing at a reasonable cost, thus alleviating ICFS.

Institutional investors' SVs can enhance monitoring and incentives for management and mitigate agency conflicts [11], thus dampening ICFS. First, conducting SVs can help enhance the monitoring and governance ability of institutional investors, thus restraining management's investment behaviors, which are detrimental to a company's long-term development [52]. Through conducting SVs, institutional investors can have face-to-face conversations with the company management, and capture information that has not yet been disclosed by the company by paying attention to details, such as the words, tone of voice, facial expressions, and body movements of the executives who receive the research promptly. They may also uncover the information concealed by management [56]. Therefore, information generated by institutional investors through SVs helps enhance the monitoring and governance ability of institutional investors and inhibits ICFS. Second, the randomness and continuity of SVs can help institutional investors form long-term, effective supervision of the companies being visited, increases the probability of discovering management's opportunistic behaviors, and thus restrains management's self-interest motivated investment behaviors [57]. Due to the elevated pressure of continuous monitoring by institutional investors and increased rent-seeking costs, management may be inclined to make investment decisions that are consistent with the interests of the principal, which will also alleviate the conflicts of interest between shareholders and management and inhibit ICFS. Third, institutional investors' SVs can reduce investment myopia by exerting market pressure on company management [58]. After conducting SVs, institutional investors' continued attention, buying of shares, increase in holdings, and release of positive research reports will send positive signals to other market participants and enhance management's reputation, whereas institutional investors' cancellation of attention, reduction of holdings, and release of negative research reports will signal to the market the existence of investment risks in the firm [54]. Therefore, to avoid threats to its professional reputation and security due to negative behaviors, such as institutional investors reducing their holdings and releasing negative research reports, management may reduce self-interested behaviors, mitigate agency conflicts, and reduce irrational investment behaviors that are detrimental to the long-term interests of the firm, thereby suppressing ICFS.

Based on the above analysis, this paper proposes the following hypothesis.

*H1*: *Ceteris paribus*, *institutional investors' SVs can reduce ICFS.*

Based on further analysis of the impact mechanisms above, this paper proposes the following hypotheses:

*H2a*: *Ceteris paribus*, *institutional investors' SVs can mitigate financing constraints and thus reduce ICFS.*

*H2b*: *Ceteris paribus*, *institutional investors' SVs can inhibit agency conflicts and thus reduce ICFS.*

## 4. Methodology

### 4.1 Sample selection and data source

In 2012, the SZSE took steps to enhance the fairness and transparency of corporate SVs and issued the China Fair Disclosure requirement, which requires companies to publish standard summary reports with detailed information via the stock exchange's web portal (Hudongyi website) within two trading days of each site visit. The information required to be disclosed includes details such as the time of the SVs, the names of the visitors and the receptionists, questions asked by visitors, and the corresponding answers. The research sample includes Chinese A-share companies listed on the SZSE from 2013 to 2018. The study excludes financial companies, ST companies, and companies with missing key variables. Table 1 presents our data selection process. The final sample includes 9,626 firm-year observations of 2,079 unique firms. Fig 1 shows the distribution of companies that host site visits within the sample period. The data used in this paper come from the China Stock Market & Accounting Research Database (CSMAR). All continuous variables are winsorized to minimize the effects of outliers.

### 4.2 Measurement model

We construct the following fixed-effects model to test whether and how institutional investors' SVs affect ICFS.

$$Invest_{i,t} = \alpha_0 + \alpha_1 \times CF_{i,t} + \alpha_2 \times inv_{i,t} + \alpha_3 \times CF_{i,t}*inv_{i,t} + \alpha_4 \times Size_{i,t} + \alpha_5 \times Lev_{i,t} + \alpha_6$$
$$\times Q_{i,t} + \alpha_7 \times Return_{i,t} + \alpha_8 \times Age_{i,t} + \alpha_9 \times Finindex_{i,t} + \Sigma Firm + \Sigma Year + \varepsilon_{i,t} \quad (1)$$

Where $\alpha_0$ is the intercept, $\alpha_{1-9}$ represents the regression coefficients, $\varepsilon_{i,t}$ is the error term, and subscripts i and t denote firm and year, respectively. $\Sigma Firm$ and $\Sigma Year$ denote firm-fixed and year-fixed effects, respectively. The dependent variable *Invest* denotes corporate investment expenditure. The independent variable *CF* denotes internal cash flow. The independent variable *inv* denotes institutional investors' SVs, while, *inv_fre* and *inv_bre* are the two ways of measuring *inv* simultaneously. Standard errors are adjusted for clustering when testing the statistical significance of the coefficient estimates. The coefficient of $CF*inv$, $\alpha_3$, in model (1), is the main parameter to be estimated, and if H1 holds, then its coefficient estimate should be significantly negative.

### 4.3 Variable definitions

**4.3.1 Dependent variable.** The dependent variable *Invest* denotes investment expenditure, measured as the ratio of cash payments for fixed assets, intangible assets, and other long-term assets minus the ratio of cash receipts from selling these assets to the beginning total assets.

**Table 1. Sample selection.**

| Sample selection process | Observations |
| --- | --- |
| A-share firms listed on the Shenzhen Stock Exchange in China during 2013–2018 | 10,909 |
| Delete: Firms in the financial industry | (103) |
| Delete: Firms with special treatment such as named ST, *ST and delisted | (311) |
| Delete: Firms with missing values | (869) |
| Final sample | 9,626 |

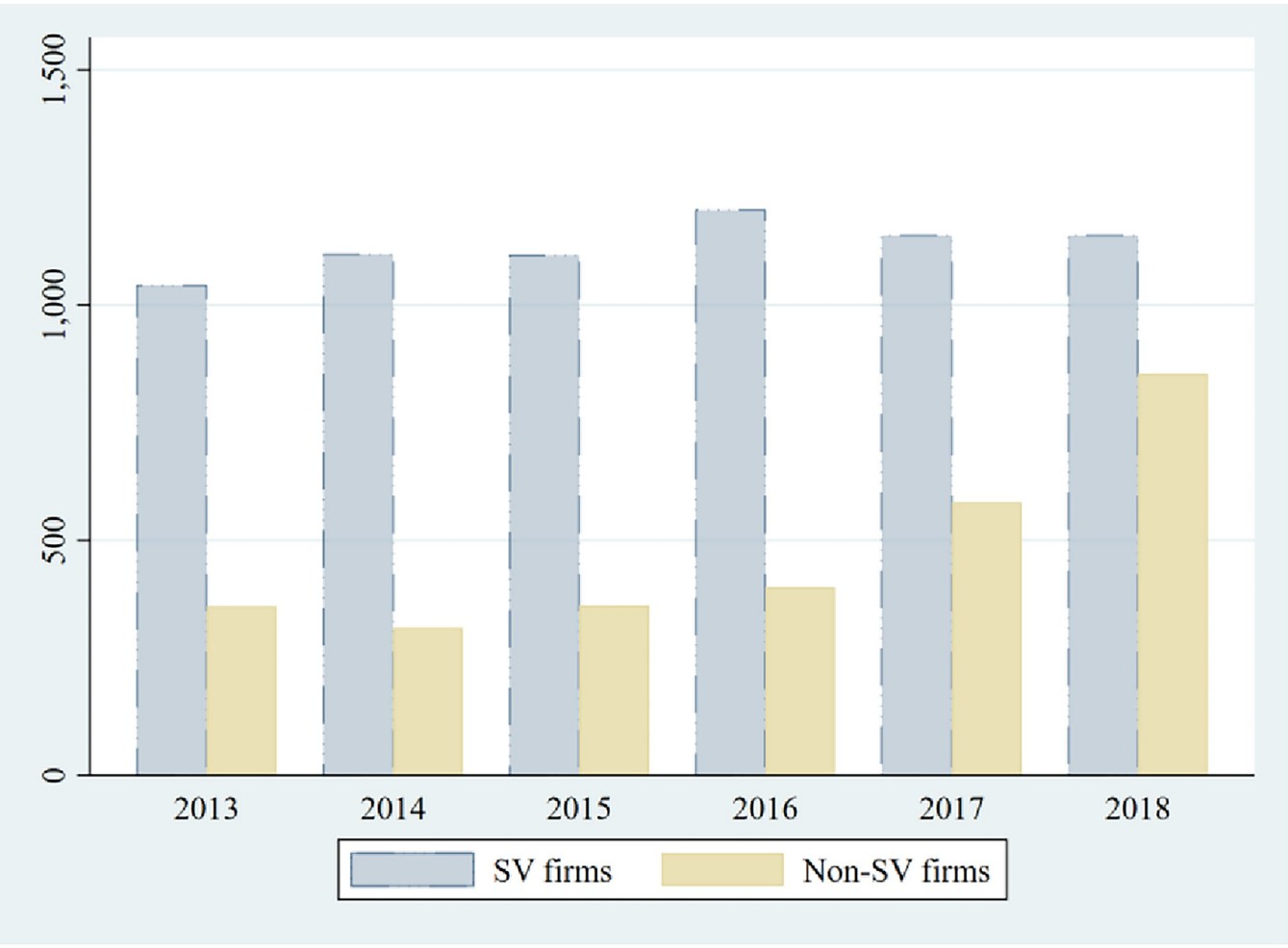

**Fig 1. Sample distribution.**

**4.3.2 Independent variables: Internal cash flow.** The independent variable *CF* denotes internal cash flow, measured as the ratio of net cash flow from operating activities to the beginning total assets.

**4.3.3 Independent variables: Institutional investors' SVs.** The independent variable *inv* denotes institutional investors' SVs. Two proxies are used to measure *inv* simultaneously, where *inv_fre* denotes the frequency of institutional investors' SVs, measured as the natural logarithm of one plus the number of SVs to a company by all institutional investors during a given year [1], and *inv_bre* denotes the breadth of institutional investors' SVs, measured as the natural logarithm of one plus the number of institutional investors that conduct SVs to a company during a given year [25]. For companies that do not disclose any information about institutional investors' SVs, *inv_fre* and *inv_bre are* set to zero [11].

**4.3.4 Control variables.** Referring to the existing literature [38, 41, 59], we control for the following variables: (1) Firm size (*Size*), equals the natural logarithm of total assets at the end of the year. (2) Leverage ratio (*Lev*), equals the ratio of total liabilities to total assets. (3) Investment opportunity (*Q*), equals the ratio of the market value of equity plus the book value of liabilities to total assets. (4) Return on assets (*Return*), equals the ratio of net profit to total assets. (5) Firm age (*Age*), equals the number of established years of the company. (6) Regional

**Table 2. Descriptive statistics.**

| Variables | N | Mean | Median | Std. Dev | Minimum value | Maximum value |
|---|---|---|---|---|---|---|
| Invest | 9626 | 0.058 | 0.038 | 0.063 | -0.020 | 0.329 |
| CF | 9626 | 0.048 | 0.046 | 0.085 | -0.228 | 0.324 |
| inv_fre | 9626 | 1.138 | 1.099 | 0.958 | 0 | 3.332 |
| inv_bre | 9626 | 2.108 | 2.303 | 1.704 | 0 | 5.371 |
| Size | 9626 | 21.950 | 21.830 | 1.097 | 19.860 | 25.280 |
| Lev | 9626 | 0.393 | 0.379 | 0.200 | 0.050 | 0.863 |
| Q | 9626 | 2.915 | 2.276 | 2.068 | 0.896 | 13.100 |
| Return | 9626 | 0.041 | 0.040 | 0.063 | -0.250 | 0.215 |
| Age | 9626 | 2.804 | 2.833 | 0.334 | 1.946 | 3.466 |
| Finindex | 9626 | 8.267 | 9.025 | 1.790 | 0.450 | 10.900 |

financial development (*Finindex*), equals the financial development index of the region where the company is registered.

## 5. Empirical analysis

### 5.1 Descriptive statistics and correlation analysis

Descriptive statistics for the variables are presented in Table 2. We can find that the mean value of *Invest* is 0.058, and the difference between the maximum and minimum values of *Invest* is 0.349, indicating that there are obvious variations in investment expenditure among the Chinese A-share companies listed on the SZSE. The mean and median values of *CF* are 0.048 and 0.046, respectively, with a difference of 0.552 between the maximum and minimum values, indicating that there are significant differences in cash flow among the different listed companies. The mean values of *inv_fre* and *inv_bre* are 1.138 and 2.108, respectively, and the maximum of *inv_fre* and *inv_bre* are 3.332 and 5.371, respectively. Both are much larger than the minimum, which means that there are considerable variations in the frequency and breadth of institutional investors' SVs among companies. Moreover, the intervals of the remaining variables are reasonable.

Pearson correlations for the main variables are presented in Table 3. It can be found that *Investe* is significantly and positively correlated with *CF* (the Pearson coefficient is 0.206),

**Table 3. Pearson correlation coefficients.**

| | Invest | CF | inv_fre | inv_bre | Size | Lev | Q | Return | Age | Finindex |
|---|---|---|---|---|---|---|---|---|---|---|
| Invest | 1 | | | | | | | | | |
| CF | 0.206*** | 1 | | | | | | | | |
| inv_fre | 0.129*** | 0.096*** | 1 | | | | | | | |
| inv_bre | 0.151*** | 0.112*** | 0.886*** | 1 | | | | | | |
| Size | 0.000 | 0.016 | 0.133*** | 0.157*** | 1 | | | | | |
| Lev | -0.027*** | -0.196*** | -0.059*** | -0.072*** | 0.530*** | 1 | | | | |
| Q | 0.045*** | 0.131*** | 0.044*** | 0.107*** | -0.492*** | -0.358*** | 1 | | | |
| Return | 0.170*** | 0.392*** | 0.245*** | 0.292*** | 0.000 | -0.339*** | 0.221*** | 1 | | |
| Age | -0.119*** | -0.026*** | -0.183*** | -0.183*** | 0.180*** | 0.167*** | -0.102*** | -0.071*** | 1 | |
| Finindex | 0.052*** | 0.037*** | 0.127*** | 0.096*** | -0.079*** | -0.023** | 0.008 | 0.043*** | -0.104*** | 1 |

Notes: $^{*}$p < 0.1

$^{**}$p < 0.05

$^{***}$p < 0.01.

which tentatively suggests that corporate cash flow has a significant positive effect on investment expenditure in the full sample. This is a well-known finding that has been documented in the existing literature [15, 22, 26, 46, 59]. In this paper, a regression analyses is required to verify the relationship between institutional investors' SVs and ICFS. In addition, the correlation coefficients between any two explanatory variables are small. We perform a variance inflation factor (VIF) test and find that the maximum value of VIF is 2.83, which implies that there is no serious problem with multicollinearity among the explanatory variables.

## 5.2 Institutional investors' SVs and corporate ICFS

Table 4 displays the regression results of the fixed-effects model. Column (1) shows a baseline of the typical investment–cash flow regression specification, in which there is a significant positive correlation between *CF* and *Invest* (the coefficient is 0.053 and significantly at the 1% level), indicating that corporate investment expenditure is largely dependent on current cash flow from operating activities. This is consistent with the existing literature [15, 22, 26, 46, 59].

**Table 4. Institutional investors' SVs and corporate ICFS.**

| | (1) | (2) | (3) |
|---|---|---|---|
| CF | 0.053*** | 0.079*** | 0.074*** |
| | (0.011) | (0.016) | (0.016) |
| inv_fre | | 0.002** | |
| | | (0.001) | |
| CF*inv_fre | | -0.025** | |
| | | (0.010) | |
| inv_ins | | | 0.001*** |
| | | | (0.001) |
| CF*inv_ins | | | -0.010* |
| | | | (0.005) |
| Size | 0.022*** | 0.021*** | 0.021*** |
| | (0.003) | (0.003) | (0.003) |
| Lev | 0.012 | 0.012 | 0.012 |
| | (0.009) | (0.009) | (0.009) |
| Q | 0.001** | 0.001** | 0.001* |
| | (0.001) | (0.001) | (0.001) |
| Return | 0.091*** | 0.090*** | 0.088*** |
| | (0.014) | (0.014) | (0.014) |
| Age | -0.055*** | -0.054*** | -0.053*** |
| | (0.019) | (0.019) | (0.019) |
| Finindex | -0.002 | -0.002 | -0.002 |
| | (0.001) | (0.001) | (0.001) |
| Firm fixed effects | YES | YES | YES |
| Year fixed effects | YES | YES | YES |
| Constant | -0.249*** | -0.243*** | -0.241*** |
| | (0.077) | (0.077) | (0.078) |
| Observations | 9626 | 9626 | 9626 |
| Adjusted $R^2$ | 0.089 | 0.091 | 0.090 |

Notes: *p < 0.1

**p < 0.05

***p < 0.01 (robust standard errors adjusted for heteroscedasticity are reported in parentheses).

Columns (2) and (3) of Table 4 show the regression results on whether and how institutional investors' SVs affect ICFS. The coefficients of *inv_fre* and *inv_bre* are both significantly positive, indicating that the companies being visited by institutional investors are highly likely to increase their investments and strongly desired to expand for better future growth. More importantly, the coefficients of *CF*inv_fre* and *CF*inv_bre* are both significantly negative, indicating that institutional investors' SVs significantly reduce ICFS, thus supporting H1.

## 5.3 Channel test of financing constraints

Existing studies have affirmed that ICFS stemming from financing constraints is more likely to exist in companies that have valuable investment opportunities but are unable to obtain external financing. In addition, ICFS is mainly due to financing constraints in small companies and agency conflicts in large companies [40, 42, 45, 46]. Therefore, investment opportunity (*Q*) and firm size (*Size*) are selected to measure the financing constraints. To test H2a, that is, the proposal that institutional investors' SVs can mitigate financing constraints and thus reduce ICFS, we divide the full sample into financially constrained companies and financially unconstrained companies according to the annual median of *Q* and *Size*. We then examine the impact of institutional investors' SVs on ICFS using these sub-samples separately. The regression results for the subsample reported in Table 5 show that the coefficients of *CF*inv_fre* and *CF*inv_bre* are statistically insignificant in companies with good investment opportunities and small sizes, indicating that institutional investors' SVs cannot reduce ICFS caused by financing constraints. Thus, H2a is not supported. More importantly, we note that the coefficients of *CF*inv_fre* and *CF*inv_bre* are significantly negative among large-size companies. It could be speculated that inhibiting agency conflicts may be an important channel through which institutional investors' SVs reduce ICFS.

## 5.4 Channel test for agency conflicts

Several studies provide evidence of agency problems associated with the use of free cash flow that result in overinvestment [48, 60]. Therefore, we select internal cash flow (*CF*) to measure agency conflicts. Considering that managerial perk consumption and controlling shareholder's tunneling behaviors are reflected in the company's other cash-to-operating activities, we also select agency costs (*COST*, which equals cash paid for other and operating activities divided by operating income) to measure agency conflicts. To further test H2b, that is, the proposal that institutional investors' SVs can inhibit agency conflicts and thus reduce ICFS, the sample is divided into companies with serious agency conflicts and companies without serious agency conflicts according to the annual median of *CF* and *COST*. Then, the impact of institutional investors' SVs on ICFS is examined using these subsamples separately. The regression results for the subsample reported in Table 6 show that the coefficients of *CF*inv_fre* and *CF*inv_bre* are significantly negative in companies with high internal cash flow and agency costs, which supports H2b that institutional investors' SVs reduce ICFS caused by agency conflicts.

## 5.5 Further analysis: The external governance role of institutional investors' SVs

Based on the conclusion that institutional investors' SVs can inhibit agency conflicts and thus reduce ICFS, we further analyze the external governance role of institutional investors' SVs and explore in depth how it relates specifically to other mechanisms of corporate governance.

**5.5.1 The perspective of internal supervision governance.** Existing studies suggest that the lower the degree of check-and-balance ownership structure, the higher the degree of separation of ownership and control, or the smaller the size of the board of directors, the weaker

**Table 5. Channel tests for financing constraints.**

| | (1) | (2) | (3) | (4) | (5) | (6) | (7) | (8) |
|---|---|---|---|---|---|---|---|---|
| | Investment opportunities | | | | Firm size | | | |
| | Good | Poor | Good | Poor | Small | large | Small | large |
| CF | 0.060** | 0.085*** | 0.053** | 0.082*** | 0.021 | 0.079*** | 0.021 | 0.077*** |
| | (0.025) | (0.024) | (0.026) | (0.025) | (0.020) | (0.024) | (0.020) | (0.024) |
| inv_fre | 0.000 | 0.001 | | | 0.001 | 0.001 | | |
| | (0.002) | (0.001) | | | (0.002) | (0.001) | | |
| CF*inv_fre | -0.013 | -0.037** | | | 0.006 | -0.030** | | |
| | (0.014) | (0.015) | | | (0.015) | (0.013) | | |
| inv_bre | | | 0.001 | 0.001 | | | 0.001 | 0.001 |
| | | | (0.001) | (0.001) | | | (0.001) | (0.001) |
| CF*inv_bre | | | -0.003 | -0.020** | | | 0.003 | -0.016** |
| | | | (0.008) | (0.009) | | | (0.008) | (0.008) |
| Size | 0.026*** | 0.026*** | 0.026*** | 0.026*** | 0.017*** | 0.025*** | 0.016*** | 0.025*** |
| | (0.004) | (0.005) | (0.005) | (0.005) | (0.005) | (0.005) | (0.005) | (0.005) |
| Lev | 0.034*** | -0.012 | 0.034*** | -0.012 | 0.022* | 0.008 | 0.023* | 0.009 |
| | (0.013) | (0.017) | (0.013) | (0.017) | (0.012) | (0.017) | (0.012) | (0.017) |
| Q | 0.001 | 0.014*** | 0.001 | 0.014*** | 0.000 | 0.004** | 0.000 | 0.004** |
| | (0.001) | (0.004) | (0.001) | (0.004) | (0.001) | (0.002) | (0.001) | (0.002) |
| Return | 0.097*** | 0.065*** | 0.095*** | 0.064*** | 0.095*** | 0.090*** | 0.094*** | 0.089*** |
| | (0.024) | (0.023) | (0.023) | (0.023) | (0.018) | (0.024) | (0.018) | (0.024) |
| Age | -0.085*** | -0.030 | -0.083*** | -0.029 | -0.037 | -0.089*** | -0.036 | -0.088*** |
| | (0.031) | (0.024) | (0.031) | (0.024) | (0.027) | (0.031) | (0.027) | (0.031) |
| Finindex | -0.004 | 0.000 | -0.004 | 0.000 | -0.002 | -0.001 | -0.002 | -0.001 |
| | (0.002) | (0.002) | (0.002) | (0.002) | (0.002) | (0.002) | (0.002) | (0.002) |
| Firm fixed effects | Yes | Yes | Yes | Yes | Yes | Yes | Yes | Yes |
| Year fixed effects | Yes | Yes | Yes | Yes | Yes | Yes | Yes | Yes |
| Constant | -0.235** | -0.458*** | -0.231** | -0.455*** | -0.174 | -0.263* | -0.165 | -0.260* |
| | (0.113) | (0.125) | (0.114) | (0.125) | (0.111) | (0.137) | (0.110) | (0.137) |
| Observations | 4813 | 4813 | 4813 | 4813 | 4813 | 4813 | 4813 | 4813 |
| Adjusted $R^2$ | 0.088 | 0.086 | 0.088 | 0.086 | 0.102 | 0.082 | 0.103 | 0.082 |

Notes: *$p < 0.1$

**$p < 0.05$

***$p < 0.01$ (robust standard errors adjusted for heteroscedasticity are reported in parentheses).

the companies' internal supervision and governance, and the more serious the principal-agent problems may be. By grouping the above-mentioned corporate governance characteristics, we examine whether the external governance mechanisms of institutional investors' SVs and the internal supervision governance are complementary or alternative in reducing ICFS. The regression results for the subsample reported in Table 7 show that the coefficients of CF*inv_fre and CF*inv_bre are significantly negative in companies with a lower degree of check-and-balance ownership structure, a higher degree of separation of ownership and control, and a smaller size of the board of directors, that is, institutional investors' SVs reduce ICFS in companies with poor internal supervision governance, indicating that institutional investors' SVs and other forms of corporate governance mechanisms operate as substitutes, rather than complements in reducing ICFS.

**Table 6. Channel tests for agency conflicts.**

| | (1) | (2) | (3) | (4) | (5) | (6) | (7) | (8) |
|---|---|---|---|---|---|---|---|---|
| | Internal cash flow | | | | Agency costs | | | |
| | Low | High | Low | High | Low | High | Low | High |
| CF | -0.018 | 0.226*** | -0.022 | 0.227*** | 0.067*** | 0.075*** | 0.056** | 0.078*** |
| | (0.019) | (0.042) | (0.019) | (0.043) | (0.022) | (0.022) | (0.023) | (0.023) |
| inv_fre | 0.002 | 0.005* | | | 0.004*** | 0.001 | | |
| | (0.001) | (0.003) | | | (0.001) | (0.001) | | |
| CF*inv_fre | -0.009 | -0.055** | | | -0.016 | -0.035*** | | |
| | (0.014) | (0.025) | | | (0.013) | (0.013) | | |
| inv_bre | | | 0.001* | 0.004** | | | 0.002*** | 0.001 |
| | | | (0.001) | (0.002) | | | (0.001) | (0.001) |
| CF*inv_bre | | | -0.003 | -0.029** | | | -0.004 | -0.019*** |
| | | | (0.008) | (0.013) | | | (0.008) | (0.007) |
| Size | 0.010*** | 0.036*** | 0.010*** | 0.035*** | 0.022*** | 0.016*** | 0.022*** | 0.016*** |
| | (0.003) | (0.005) | (0.003) | (0.005) | (0.005) | (0.004) | (0.005) | (0.004) |
| Lev | -0.006 | 0.033** | -0.006 | 0.033** | 0.011 | 0.017 | 0.010 | 0.017 |
| | (0.011) | (0.016) | (0.011) | (0.016) | (0.015) | (0.012) | (0.015) | (0.012) |
| Q | 0.001 | 0.001 | 0.001 | 0.001 | 0.001 | 0.000 | 0.001 | 0.000 |
| | (0.001) | (0.001) | (0.001) | (0.001) | (0.001) | (0.001) | (0.001) | (0.001) |
| Return | 0.075*** | 0.095*** | 0.074*** | 0.092*** | 0.098*** | 0.086*** | 0.095*** | 0.086*** |
| | (0.017) | (0.028) | (0.017) | (0.027) | (0.022) | (0.020) | (0.022) | (0.020) |
| Age | -0.044** | -0.032 | -0.044** | -0.030 | -0.082*** | -0.016 | -0.082*** | -0.015 |
| | (0.022) | (0.028) | (0.022) | (0.028) | (0.029) | (0.025) | (0.029) | (0.025) |
| Finindex | -0.002 | -0.002 | -0.002 | -0.002 | -0.003 | 0.000 | -0.003 | 0.000 |
| | (0.002) | (0.002) | (0.002) | (0.002) | (0.002) | (0.002) | (0.002) | (0.002) |
| Firm fixed effects | Yes | Yes | Yes | Yes | Yes | Yes | Yes | Yes |
| Year fixed effects | Yes | Yes | Yes | Yes | Yes | Yes | Yes | Yes |
| Constant | -0.027 | -0.625*** | -0.021 | -0.621*** | -0.183 | -0.246** | -0.176 | -0.244** |
| | (0.087) | (0.115) | (0.087) | (0.116) | (0.125) | (0.101) | (0.126) | (0.101) |
| Observations | 4813 | 4813 | 4813 | 4813 | 4812 | 4813 | 4812 | 4813 |
| Adjusted $R^2$ | 0.085 | 0.126 | 0.086 | 0.126 | 0.089 | 0.095 | 0.089 | 0.095 |

Notes: p < 0.1; **p < 0.05; ***p < 0.01 (robust standard errors adjusted for heteroscedasticity are reported in parentheses).

**5.5.2 The perspective of senior executives' compensation incentive mechanisms.** In addition to internal supervision governance, senior executives' compensation incentive mechanisms are a key element of corporate governance, and agency conflicts resulting from the opportunistic motives of senior executives may be more severe when their compensation incentive mechanisms are inadequate. We select senior executives' monetary remuneration (*Salary1*) and equity remuneration (*Salary2*) to measure senior executives' compensation incentive mechanisms. To further examine whether institutional investors' SVs and senior executives' compensation incentive mechanisms are complementary or alternative to reducing ICFS, the full sample is divided into two subsamples according to the annual median of *Salary1* and *Salary2*. Then, the impact of institutional investors' SVs on ICFS is examined using these subsamples separately. The regression results for the subsample reported in Table 8 show that the coefficients of *CF*inv_fre* and *CF*inv_bre* are significantly negative in companies with a lower monetary remuneration and equity remuneration of senior executives, that is, institutional investors' SVs reduce ICFS in companies with weak compensation incentive

**Table 7. The perspective of internal supervision governance.**

| | (1) | (2) | (3) | (4) | (5) | (6) | (7) | (8) | (9) | (10) | (11) | (12) |
|---|---|---|---|---|---|---|---|---|---|---|---|---|
| | Degree of check-and-balance ownership structure | | | | Degree of separation of ownership and control | | | | Size of the board of directors | | | |
| | Low | High | Low | High | High | Low | High | Low | Small | large | Small | large |
| CF | 0.067*** | 0.072*** | 0.066*** | 0.067*** | 0.082*** | 0.056** | 0.073*** | 0.054** | 0.081*** | 0.073*** | 0.084*** | 0.058** |
| | (0.022) | (0.023) | (0.023) | (0.024) | (0.024) | (0.023) | (0.024) | (0.023) | (0.023) | (0.025) | (0.023) | (0.026) |
| inv_fre | 0.001 | 0.001 | | | 0.002 | 0.002 | | | 0.003** | 0.000 | | |
| | (0.001) | (0.001) | | | (0.001) | (0.001) | | | (0.001) | (0.001) | | |
| CF*inv_fre | -0.029** | -0.018 | | | -0.042*** | 0.000 | | | -0.026* | -0.024 | | |
| | (0.014) | (0.014) | | | (0.014) | (0.015) | | | (0.015) | (0.015) | | |
| inv_bre | | | 0.000 | 0.001* | | | 0.002** | 0.001* | | | 0.002*** | 0.000 |
| | | | (0.001) | (0.001) | | | (0.001) | (0.001) | | | (0.001) | (0.001) |
| CF*inv_bre | | | -0.015** | -0.007 | | | -0.017** | 0.001 | | | -0.016** | -0.005 |
| | | | (0.007) | (0.008) | | | (0.008) | (0.008) | | | (0.008) | (0.009) |
| Size | 0.023*** | 0.018*** | 0.023*** | 0.018*** | 0.025*** | 0.020*** | 0.024*** | 0.020*** | 0.014*** | 0.027*** | 0.014*** | 0.027*** |
| | (0.004) | (0.004) | (0.004) | (0.004) | (0.004) | (0.005) | (0.004) | (0.005) | (0.004) | (0.004) | (0.004) | (0.004) |
| Lev | 0.009 | 0.034*** | 0.009 | 0.035*** | -0.001 | 0.035** | -0.001 | 0.035** | 0.019 | -0.000 | 0.019 | -0.001 |
| | (0.015) | (0.013) | (0.015) | (0.013) | (0.013) | (0.015) | (0.013) | (0.015) | (0.013) | (0.014) | (0.013) | (0.014) |
| Q | 0.002* | 0.001 | 0.002** | 0.000 | 0.002** | 0.000 | 0.002** | 0.000 | 0.000 | 0.003** | 0.000 | 0.003** |
| | (0.001) | (0.001) | (0.001) | (0.001) | (0.001) | (0.001) | (0.001) | (0.001) | (0.001) | (0.001) | (0.001) | (0.001) |
| Return | 0.098*** | 0.094*** | 0.099*** | 0.092*** | 0.057*** | 0.120*** | 0.054*** | 0.119*** | 0.116*** | 0.038 | 0.115*** | 0.036 |
| | (0.019) | (0.020) | (0.019) | (0.020) | (0.021) | (0.020) | (0.020) | (0.020) | (0.019) | (0.024) | (0.019) | (0.023) |
| Age | -0.066** | -0.011 | -0.065** | -0.009 | -0.087*** | -0.025 | -0.085*** | -0.025 | -0.047* | -0.077*** | -0.047* | -0.077*** |
| | (0.027) | (0.026) | (0.027) | (0.026) | (0.030) | (0.025) | (0.030) | (0.025) | (0.027) | (0.027) | (0.027) | (0.027) |
| Finindex | -0.004* | 0.000 | -0.004* | 0.000 | -0.002 | -0.003 | -0.002 | -0.003 | -0.003 | -0.000 | -0.003 | -0.000 |
| | (0.002) | (0.002) | (0.002) | (0.002) | (0.002) | (0.002) | (0.002) | (0.002) | (0.003) | (0.002) | (0.003) | (0.002) |
| Firm fixed effects | Yes | Yes | Yes | Yes | Yes | Yes | Yes | Yes | Yes | Yes | Yes | Yes |
| Year fixed effects | Yes | Yes | Yes | Yes | Yes | Yes | Yes | Yes | Yes | Yes | Yes | Yes |
| Constant | -0.242** | -0.313*** | -0.247** | -0.305*** | -0.228* | -0.286** | -0.230* | -0.277** | -0.095 | -0.312*** | -0.088 | -0.315*** |
| | (0.117) | (0.102) | (0.118) | (0.103) | (0.118) | (0.113) | (0.119) | (0.113) | (0.106) | (0.109) | (0.106) | (0.110) |
| Observations | 4813 | 4813 | 4813 | 4813 | 4668 | 4668 | 4668 | 4668 | 4813 | 4813 | 4813 | 4813 |
| Adjusted $R^2$ | 0.073 | 0.104 | 0.073 | 0.104 | 0.075 | 0.105 | 0.074 | 0.105 | 0.088 | 0.080 | 0.089 | 0.079 |

Notes: p < 0.1; **p < 0.05; ***p < 0.01 (robust standard errors adjusted for heteroscedasticity are reported in parentheses).

mechanisms. This indicates that there is a reciprocal substitution relationship between institutional investors' SVs and senior executives' compensation incentive mechanisms in reducing ICFS.

## 5.6 Robustness test

**5.6.1 Propensity Score Matching (PSM).** Institutional investors' SVs are deliberate choices based on a firm's characteristics. To mitigate the problem of selection bias, we employ the propensity score matching (PSM) approach. The details of the PSM procedure are reported in the S2 Appendix. We re-run the regression using the matched sample, and our main results hold after controlling for potential self-selection bias, as shown in Table 9.

**5.6.2 Two-stage least squares regressions.** The findings of this paper may be affected by endogeneity. For example, institutional investors may choose to conduct SVs to companies for unobservable reasons. Thus, omitted variables may cause bias in the results obtained in this

**Table 8. The perspective of senior executives' compensation incentive mechanisms.**

| | (1) | (2) | (3) | (4) | (5) | (6) | (7) | (8) |
|---|---|---|---|---|---|---|---|---|
| | Senior executives' monetary remuneration | | | | Senior executives' equity remuneration | | | |
| | Low | High | Low | High | Yes | No | Yes | No |
| CF | 0.095*** | 0.050** | 0.090*** | 0.040* | 0.092*** | 0.050** | 0.085*** | 0.045* |
| | (0.021) | (0.024) | (0.021) | (0.025) | (0.024) | (0.023) | (0.023) | (0.024) |
| inv_fre | 0.003** | 0.000 | | | 0.000 | 0.001 | | |
| | (0.002) | (0.001) | | | (0.001) | (0.001) | | |
| CF*inv_fre | -0.038*** | -0.011 | | | -0.034** | 0.002 | | |
| | (0.014) | (0.013) | | | (0.015) | (0.014) | | |
| inv_bre | | | 0.002* | 0.001 | | | 0.001 | 0.001 |
| | | | (0.001) | (0.001) | | | (0.001) | (0.001) |
| CF*inv_bre | | | -0.018** | -0.002 | | | -0.014* | 0.003 |
| | | | (0.008) | (0.008) | | | (0.008) | (0.008) |
| Size | 0.024*** | 0.023*** | 0.024*** | 0.022*** | 0.025*** | 0.020*** | 0.024*** | 0.020*** |
| | (0.005) | (0.005) | (0.005) | (0.005) | (0.004) | (0.005) | (0.004) | (0.005) |
| Lev | -0.001 | 0.034** | -0.001 | 0.035** | 0.004 | 0.036** | 0.004 | 0.036** |
| | (0.013) | (0.015) | (0.013) | (0.015) | (0.014) | (0.015) | (0.014) | (0.015) |
| Q | 0.000 | 0.003*** | 0.000 | 0.003*** | 0.002* | 0.000 | 0.002* | 0.000 |
| | (0.001) | (0.001) | (0.001) | (0.001) | (0.001) | (0.001) | (0.001) | (0.001) |
| Return | 0.061*** | 0.106*** | 0.061*** | 0.102*** | 0.053*** | 0.113*** | 0.050*** | 0.113*** |
| | (0.019) | (0.022) | (0.019) | (0.021) | (0.019) | (0.019) | (0.019) | (0.018) |
| Age | -0.075** | -0.050* | -0.073** | -0.049* | -0.103*** | 0.008 | -0.101*** | 0.008 |
| | (0.030) | (0.027) | (0.030) | (0.027) | (0.029) | (0.027) | (0.029) | (0.027) |
| Finindex | -0.001 | -0.003 | -0.001 | -0.003 | -0.003 | 0.001 | -0.003 | 0.001 |
| | (0.002) | (0.003) | (0.002) | (0.003) | (0.002) | (0.002) | (0.002) | (0.002) |
| Firm fixed effects | Yes | Yes | Yes | Yes | Yes | Yes | Yes | Yes |
| Year fixed effects | Yes | Yes | Yes | Yes | Yes | Yes | Yes | Yes |
| Constant | -0.239** | -0.299** | -0.236** | -0.293** | -0.171 | -0.401*** | -0.175 | -0.397*** |
| | (0.119) | (0.122) | (0.119) | (0.123) | (0.108) | (0.127) | (0.108) | (0.128) |
| Observations | 4783 | 4837 | 4783 | 4837 | 4739 | 4803 | 4739 | 4803 |
| Adjusted $R^2$ | 0.100 | 0.076 | 0.099 | 0.076 | 0.082 | 0.102 | 0.080 | 0.102 |

Notes: $p < 0.1$; **$p < 0.05$; ***$p < 0.01$ (robust standard errors adjusted for heteroscedasticity are reported in parentheses).

paper. In addition, ICFS is likely to be an important reference for institutional investors in deciding whether to conduct SVs to companies or not, which in turn affects institutional investors' SVs. To mitigate the endogeneity problem, following the existing literature [11, 61, 62], we conduct a two-stage least squares (2SLS) regression using the dummy variable of China Securities Index 300 (CSI 300) index constituents (*IV1*) and the mean value of *inv_fre* in the same city in the same year (*IV2*) as instrumental variables.

The validity tests of the instrumental variables show that the Kleibergen-Paap rk LM statistic was 472.446 and 392.274, respectively, which are significant at the 1% level, rejecting the hypothesis that "under-recognition". The Cragg-Donald Wald F statistics are 224.393 and 182.725, respectively, which rejects the hypothesis that "weak instrumental variables". The Hansen J statistic corresponds to a p-value of 0.2490 and 0.1502 respectively, which fails to reject the hypothesis that "all instrumental variables are exogenous". Those results indicate that both *IV1* and *IV2* are valid instrumental variables. From the second-stage estimation results in Table 10, it is inferred that a significant positive correlation exists between *CF* and *Invest*, and

**Table 9. Propensity Score Matching method (PSM).**

|  | (1) | (2) | (3) |
|---|---|---|---|
| CF | 0.052*** | 0.087*** | 0.080*** |
|  | (0.011) | (0.018) | (0.019) |
| inv_fre |  | 0.002* |  |
|  |  | (0.001) |  |
| CF*inv_fre |  | -0.030*** |  |
|  |  | (0.011) |  |
| inv_bre |  |  | 0.001** |
|  |  |  | (0.001) |
| CF*inv_bre |  |  | -0.013** |
|  |  |  | (0.006) |
| Size | 0.024*** | 0.023*** | 0.023*** |
|  | (0.003) | (0.003) | (0.003) |
| Lev | 0.015 | 0.015 | 0.015 |
|  | (0.010) | (0.010) | (0.010) |
| Q | 0.001** | 0.001** | 0.001* |
|  | (0.001) | (0.001) | (0.001) |
| Return | 0.108*** | 0.108*** | 0.107*** |
|  | (0.017) | (0.017) | (0.017) |
| Age | -0.048** | -0.049** | -0.047** |
|  | (0.020) | (0.020) | (0.020) |
| Finindex | -0.002 | -0.002 | -0.002 |
|  | (0.002) | (0.002) | (0.002) |
| Firm fixed effects | Yes | Yes | Yes |
| Year fixed effects | Yes | Yes | Yes |
| Constant | -0.308*** | -0.301*** | -0.300*** |
|  | (0.084) | (0.084) | (0.084) |
| Observations | 8681 | 8681 | 8681 |
| Adjusted $R^2$ | 0.090 | 0.092 | 0.092 |

Notes: *p < 0.1

**p < 0.05

***p < 0.01 (robust standard errors adjusted for heteroscedasticity are reported in parentheses).

the coefficients of CF*inv_fre and CF*inv_bre are both significantly negative, which means that the conclusion of this paper is still valid after correcting for endogeneity bias.

**5.6.3 Controlling for the industry fixed effect based on the ordinary least square (OLS) regression.** In the previous paragraph, the fixed effects model is used to estimate the model (1) to eliminate constant omitted variable bias. Here, model (1) is re-estimated after controlling for the industry fixed effect based on the ordinary least square (OLS) regression. The conclusion that institutional investors' SVs can reduce ICFS still holds, as shown in Table 11.

**5.6.4 Controlling for the previous year's investment expenditure.** We further control for the previous year's investment expenditure (L1.Invest) in the regression model (1) as a robustness test. The results in Table 12 support the conclusion that institutional investors' SVs can reduce ICFS.

# 6. Conclusions and policy implications

Using the unique datasets from Chinese non-financial A-share companies listed on the SZSE between 2013 and 2018, we find that overall, corporate investment expenditure is largely

**Table 10. Two-stage least squares regressions.**

|  | (1) | (2) |
|---|---|---|
| CF | 0.114*** | 0.115*** |
|  | (0.027) | (0.028) |
| inv_fre | 0.007*** |  |
|  | (0.003) |  |
| CF*inv_fre | -0.057*** |  |
|  | (0.022) |  |
| inv_ins |  | 0.004** |
|  |  | (0.002) |
| CF*inv_ins |  | -0.031** |
|  |  | (0.012) |
| Size | 0.020*** | 0.020*** |
|  | (0.002) | (0.003) |
| Lev | 0.014* | 0.014* |
|  | (0.008) | (0.008) |
| Q | 0.001* | 0.001 |
|  | (0.001) | (0.001) |
| Return | 0.083*** | 0.083*** |
|  | (0.014) | (0.014) |
| Age | -0.050*** | -0.050*** |
|  | (0.015) | (0.015) |
| Finindex | -0.002 | -0.002 |
|  | (0.001) | (0.001) |
| Firm fixed effects | YES | YES |
| Year fixed effects | YES | YES |
| Observations | 9423 | 9423 |
| $R^2$ | 0.087 | 0.087 |
| Kleibergen-Paap rk LM statistic | 472.446*** | 392.274*** |
| Cragg-Donald Wald F statistic | 224.393 | 182.725 |
| Hansen J statistic | 2.781 | 3.792 |
| p-value | 0.2490 | 0.1502 |

Notes: *$p < 0.1$

**$p < 0.05$

***$p < 0.01$ (robust standard errors adjusted for heteroscedasticity are reported in parentheses).

dependent on current cash flow from operating activities, and institutional investors' SVs can effectively reduce ICFS. These results remain robust even after employing the fixed-effects model and the PSM approach to mitigate potential endogenous problems and conduct other robustness tests. The results of channel tests show that institutional investors' SVs have a more significant inhibitory effect on ICFS in the subsamples with poor investment opportunities, large firm size, high internal cash flows, and high agency costs, demonstrating that institutional investors' SVs can reduce ICFS caused by agency conflicts rather than financing constraints. Additionally and more importantly, the disincentive effect of institutional investors' SVs on ICFS is mainly found in companies with poor internal supervision governance and weak executive compensation incentive mechanisms, indicating that institutional investors' SVs and other forms of corporate governance mechanisms operate as substitutes, rather than complements in reducing ICFS.

**Table 11. Controlling for the industry fixed effect based on the OLS regression.**

|  | (1) | (2) | (3) |
|---|---|---|---|
| CF | 0.106*** | 0.127*** | 0.127*** |
|  | (0.010) | (0.015) | (0.015) |
| inv_fre |  | 0.006*** |  |
|  |  | (0.001) |  |
| CF*inv_fre |  | -0.019** |  |
|  |  | (0.009) |  |
| inv_bre |  |  | 0.004*** |
|  |  |  | (0.000) |
| CF*inv_bre |  |  | -0.010* |
|  |  |  | (0.005) |
| Size | 0.001 | 0.000 | -0.001 |
|  | (0.001) | (0.001) | (0.001) |
| Lev | 0.034*** | 0.036*** | 0.036*** |
|  | (0.004) | (0.004) | (0.004) |
| Q | 0.001*** | 0.001*** | 0.001** |
|  | (0.000) | (0.000) | (0.000) |
| Return | 0.116*** | 0.103*** | 0.094*** |
|  | (0.010) | (0.010) | (0.010) |
| Age | -0.014*** | -0.013*** | -0.012*** |
|  | (0.002) | (0.002) | (0.002) |
| Finindex | 0.001*** | 0.001*** | 0.001** |
|  | (0.000) | (0.000) | (0.000) |
| Industry fixed effects | Yes | Yes | Yes |
| Year fixed effects | Yes | Yes | Yes |
| Constant | 0.059*** | 0.075*** | 0.091*** |
|  | (0.020) | (0.020) | (0.020) |
| Observations | 9626 | 9626 | 9626 |
| Adjusted $R^2$ | 0.106 | 0.110 | 0.114 |

Notes: *p < 0.1

**p < 0.05

***p < 0.01 (robust standard errors adjusted for heteroscedasticity are reported in parentheses).

This study has several implications. First, the information obtained by institutional investors' SVs is a useful supplement to public information that is difficult to understand and judge. When regulating the operation and management of listed companies and improving the efficiency of corporate resource allocations, the relevant government regulatory authorities should pay attention to the essential role of institutional investors' SVs. The government should also provide solid policies to guide institutional investors in conducting SVs on listed companies. This includes, strengthening the guidance and standardizing the management of investors' SVs, regulating the authenticity and reliability of the information of listed companies being visited, and improving the construction of timeliness, completeness, and standardization of the public disclosure of information on investor activity. Second, institutional investors have strong capital strength, rich investment experience, and professional talent teams, they can obtain more useful and critical information about the company's operations, corporate governance, and sustainable development through SVs. Therefore, high-quality listed companies should take the initiative to increase their willingness to interact and communicate with

**Table 12. Controlling for the previous year's investment expenditure.**

|  | (1) | (2) | (3) |
|---|---|---|---|
| CF | 0.053*** | 0.081*** | 0.074*** |
|  | (0.012) | (0.017) | (0.018) |
| inv_fre |  | 0.002* |  |
|  |  | (0.001) |  |
| CF*inv_fre |  | -0.027*** |  |
|  |  | (0.010) |  |
| inv_bre |  |  | 0.001** |
|  |  |  | (0.001) |
| CF*inv_bre |  |  | -0.011* |
|  |  |  | (0.006) |
| Size | 0.024*** | 0.024*** | 0.024*** |
|  | (0.003) | (0.003) | (0.003) |
| Lev | 0.020* | 0.021* | 0.021* |
|  | (0.011) | (0.011) | (0.011) |
| Q | 0.001* | 0.001 | 0.001 |
|  | (0.001) | (0.001) | (0.001) |
| Return | 0.081*** | 0.080*** | 0.078*** |
|  | (0.016) | (0.016) | (0.016) |
| Age | -0.010 | -0.011 | -0.010 |
|  | (0.022) | (0.022) | (0.022) |
| Finindex | -0.001 | -0.001 | -0.001 |
|  | (0.002) | (0.001) | (0.001) |
| L1.Invest | 0.079*** | 0.078*** | 0.078*** |
|  | (0.021) | (0.021) | (0.021) |
| Firm fixed effects | Yes | Yes | Yes |
| Year fixed effects | Yes | Yes | Yes |
| Constant | -0.445*** | -0.438*** | -0.436*** |
|  | (0.092) | (0.092) | (0.092) |
| Observations | 7198 | 7198 | 7198 |
| Adjusted $R^2$ | 0.087 | 0.089 | 0.088 |

Notes: *$p < 0.1$

**$p < 0.05$

***$p < 0.01$ (robust standard errors adjusted for heteroscedasticity are reported in parentheses).

external stakeholders, and make full use of the opportunity of these visits to showcase their good corporate side to avoid being undervalued by the market. More importantly, our findings may encourage listed companies with poor internal governance to discipline themselves, regulate their operations and management activities more strictly, improve their internal governance mechanisms more actively, and curb management's opportunistic behaviors at the source to avoid major problems that may lead to a series of adverse economic consequences during institutional investors' SVs.

This study highlights the importance of private communication between institutional investors and corporate managers. Despite its contributions, this study has limitations stemming from its empirical background. Our research setting is based in China, which provides an ideal and unique laboratory for studying the impact of institutional investors' SVs on ICFS. However, China's institutional context also limits the generalizability of our results. Given the

differences in national policies, institutional contexts, and legal environments between emerging and developed economies, and between China and other emerging markets, caution must be taken in applying these findings to other institutional settings. However, the objective of this study can be studied in other institutional settings. For example, future research can use international data to examine how market developments and information frictions affect the impact of investors' information-gathering activities on important corporate decisions. In addition, future research may discuss the relationship between institutional investors' SVs and other aspects of corporate activities, such as labor investment efficiency and OFDI decisions.

## Supporting information

**S1 Data.**
(ZIP)

**S1 Appendix. Variable definitions.**
(DOCX)

**S2 Appendix. The details of the PSM procedure.**
(DOCX)

## Author Contributions

**Conceptualization:** Jia Liao.

**Data curation:** Jia Liao.

**Methodology:** Yun Zhan.

**Writing – original draft:** Jia Liao, Yu Yuan.

**Writing – review & editing:** Yun Zhan.

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
