## [Decision Letter · Decision Letter 0]

28 Nov 2023

PONE-D-23-22387Institutional Investors' Site Visits and Investment-Cash Flow Sensitivity: Mitigating Financing Constraints or Inhibiting Agent Conflicts?PLOS ONE

Dear Dr. Zhan,

Thank you for submitting your manuscript to PLOS ONE. After careful consideration, we feel that it has merit but does not fully meet PLOS ONE’s publication criteria as it currently stands. Therefore, we invite you to submit a revised version of the manuscript that addresses the points raised during the review process.

**ACADEMIC EDITOR: Major revision is required as per the reviewer reports.**EDITOR REMARKS: 1. As indicated by the reviewer, we request the author to engage with a professional English editor to proofread the article and supply us with the anonymous certificate of proofreading as an evidence.2. Please highlight all changes in the text and prepare a clear table of correction.==============================

We look forward to receiving your revised manuscript.

Kind regards,

Jasman Tuyon, Ph.D., MBA

Academic Editor

PLOS ONE

Additional Editor Comments:

EDITOR REMARKS:

1. As indicated by the reviewer, we request the author to engage with a professional English editor to proofread the article and supply us with the anonymous certificate of proofreading as an evidence.

2. Please highlight all changes in the text and prepare a clear table of correction.

Reviewers' comments:

Reviewer's Responses to Questions

**Comments to the Author**

1. Is the manuscript technically sound, and do the data support the conclusions?

Reviewer #1: Partly

Reviewer #2: Yes

2. Has the statistical analysis been performed appropriately and rigorously? 

Reviewer #1: Yes

Reviewer #2: Yes

3. Have the authors made all data underlying the findings in their manuscript fully available?

Reviewer #1: Yes

Reviewer #2: Yes

4. Is the manuscript presented in an intelligible fashion and written in standard English?

Reviewer #1: Yes

Reviewer #2: No

5. Review Comments to the Author

Reviewer #1: In my opinion, it's an interesting work. However, the current version needs to get improved. In particular, the empirical analysis is not rigorous and far from complete. I have a few comments for the authors to improve the paper as summarized below:

1. The current empirical analysis is not coherent. The firm fixed effect model should be applied throughout the paper with industry fixed effect as a robustness check. Also, I suggest the authors use the IV approach to deal with endogeneity concerns. The authors could follow Jiang and Yuan (2018) and Lai, Li, Liu, and Wang (2022) to construct the instrumental variables. The details of the PSM procedure should be reported in the appendix.

2. The writing and formatting of this paper should be improved. A detailed caption of tables should be added. The authors should also provide a more detailed variable description.

Reference

Jiang, X., Yuan, Q., 2018. Institutional investors' corporate site visits and corporate innovation.

Journal of Corporate Finance, 48: 148-68.

Lai, S., Li, X., Liu, S., Wang, Q.S., 2022. Institutional investors? site visits and corporate employment decision-making. Journal of Contemporary Accounting and Economics, 18(3), 100332.

Reviewer #2: - The paper needs English proof reading.

- Tables must be organized in a better format.

- References must be checked.

- The variables table must be checked. Inv(fre) and Inv(bre) have the same definition.

- The most important problem in the research design is that the authors assumed that if no information is disclosed about SV, they assumed as zero. There must be a clear justification for this assumption.

- There must be a table explaining the sample history (initial, missing, deleted, out of scope firms etc.)

- More explanation is necessary about the theoretical background, how SV may or may not effect ICFS.

- The limitations of the study must be added.

6. PLOS authors have the option to publish the peer review history of their article (what does this mean?). If published, this will include your full peer review and any attached files.

Reviewer #1: No

Reviewer #2: No

---

## [Author Response · Author response to Decision Letter 0]

29 Jan 2024

Response to Reviewers

Dear Editor:

Thank you very much for your letter and the constructive comments from the referees about our paper entitled "Institutional Investors' Site Visits and Investment-Cash Flow Sensitivity: Mitigating Financing Constraints or Inhibiting Agent Conflicts?" (Manuscript ID: PONE-D-23-22387). In the revised manuscript of our paper, we made corresponding revisions based on the review comments (the modified parts are marked in red). The following is the modification description attached to the revised manuscript of the paper:

Response to Referee(s): 

Reviewer #1: In my opinion, it's an interesting work. However, the current version needs to get improved. In particular, the empirical analysis is not rigorous and far from complete. I have a few comments for the authors to improve the paper as summarized below:

1. The current empirical analysis is not coherent. The firm fixed effect model should be applied throughout the paper with industry fixed effect as a robustness check. Also, I suggest the authors use the IV approach to deal with endogeneity concerns. The authors could follow Jiang and Yuan (2018) and Lai, Li, Liu, and Wang (2022) to construct the instrumental variables. The details of the PSM procedure should be reported in the appendix.

Reference

Jiang, X., Yuan, Q., 2018. Institutional investors' corporate site visits and corporate innovation. Journal of Corporate Finance, 48: 148-68.

Lai, S., Li, X., Liu, S., Wang, Q.S., 2022. Institutional investors? site visits and corporate employment decision-making. Journal of Contemporary Accounting and Economics, 18(3), 100332.

Response: 

Thank you very much for your recognition and valuable suggestions for this manuscript. Based on your suggestions, we make the following revisions in the revised manuscript.

First, we have used the firm fixed effect model as the benchmark model (page 10). and carried out a series of empirical tests based on it, while using industry fixed effect (page 17) as the robustness test.

Second, we have adopted the two-stage instrumental variable method for endogeneity treatment in section 5.6.2 (page 16). The details added are as follows:

5.6.2 Two-stage least squares regressions

The findings of this paper may be affected by endogeneity. For example, institutional investors may choose to conduct SVs to companies for unobservable reasons. Thus, omitted variables may cause bias in the results obtained in this paper. In addition, ICFS is likely to be an important reference for institutional investors in deciding whether to conduct SVs to companies or not, which in turn affects institutional investors' SVs. To mitigate the endogeneity problem, following the existing literature (Jiang and Yuan 2018; Yang and Ma 2020; Lai et al. 2022), we conduct a two-stage least squares (2SLS) regression using the dummy variable of China Securities Index 300 (CSI 300) index constituents (IV1) and the mean value of inv_fre in the same city in the same year (IV2) as instrumental variables.

The validity tests of the instrumental variables show that the Kleibergen-Paap rk LM statistic was 472.446 and 392.274, respectively, which are significant at the 1% level, rejecting the hypothesis that "under-recognition". The Cragg-Donald Wald F statistics are 224.393 and 182.725, respectively, which rejects the hypothesis that "weak instrumental variables". The Hansen J statistic corresponds to a p-value of 0.2490 and 0.1502 respectively, which fails to reject the hypothesis that "all instrumental variables are exogenous". Those results indicate that both IV1 and IV2 are valid instrumental variables. From the second-stage estimation results in Table 10, it is inferred that a significant positive correlation exists between CF and Invest, and the coefficients of CF*inv_fre and CF*inv_bre are both significantly negative, which means that the conclusion of this paper is still valid after correcting for endogeneity bias.

Third, in the appendix B of the revised manuscript, we have supplemented the details of the PSM procedure (pages 36-37).

The referenced studies above are as follows:

Jiang, X., and Q. Yuan. 2018. Institutional investors' corporate site visits and corporate innovation. Journal of Corporate Finance 48:148-168.

Lai, S., X. Li, S. Liu, and Q. S. Wang. 2022. Institutional investors’ site visits and corporate employment decision-making. Journal of Contemporary Accounting & Economics 18 (3):100332.

Yang, X., and Z. Ma. 2020. Institutional Investors' Corporate Site Visits and Its Effect on Restricting the Tunneling Behavior of Large Shareholders. Journal of Central University of Finance & Economics (04):42-64.

2. The writing and formatting of this paper should be improved. A detailed caption of tables should be added. The authors should also provide a more detailed variable description.

Response: 

Thank you for your careful review and suggestions. First, this manuscript has been polished by professional institution, Scribendi, and the certificate is presented in the file Response to Reviewers R1.docx. In addition, we have checked all the changes one by one after the proofreading service to improve the linguistic quality and ensure the format accuracy of the manuscript. Second, we have added the detailed caption of tables in the main text of the revised manuscript. Third, in section 4.3 Variable definitions (pages 10-11), we have added detailed descriptions of the variables and moved the Table A Variable definitions to the appendix A(pages 35-36).

Thank you once again for your meticulous review and constructive suggestions. 

Yours Sincerely, 

The authors

Reviewer #2:

1. The paper needs English proof reading.

Response: 

Thank you for your careful review and suggestions for the proofreading of the manuscript. This manuscript has been polished by professional institution, Scribendi, and the certificate is presented in the file Response to Reviewers R1.docx. In addition, we have checked all the changes one by one after the proofreading service to improve the linguistic quality and ensure the format accuracy of the manuscript.

2. Tables must be organized in a better format.

Response: 

Thank you for your pertinent suggestions. In the revised manuscript, we present each table in a more appropriate and consistent format to clearly and visually present our empirical results. Please refer to Tables and Figures (Pages 23-35) for details.

3. References must be checked.

Response: 

Thank you for your careful review and suggestions for the references. The mistakes in references have been corrected in the revised manuscript. In addition, we have proofread repeatedly to ensure the format accuracy of the references. Please refer to Reference (Pages 20-23) for details.

4. The variables table must be checked. Inv(fre) and Inv(bre) have the same definition.

Response: 

Thank you for your careful review for the variables. In the original manuscript, the definition of inv_bre in the "Table 1 Variable definitions" was incorrect, and the definition of inv_fre was not written in sufficient detail. In the revised manuscript, we have added detailed descriptions of the variables in section 4.3.3 (Pages 10-11)and moved the Table A Variable definitions to the appendix A (Pages 35-36). 

5. The most important problem in the research design is that the authors assumed that if no information is disclosed about SV, they assumed as zero. There must be a clear justification for this assumption.

Response: 

Thank you for your valuable suggestions for the research design. In Jiang and Yuan (2018), it is noted that "For firms that do not disclose any information about institutional investors' site visits, SV is set to zero.", which is referenced in our manuscript. Although much of the literature does not emphasize it as such, the descriptive statistics of the variables in almost all the relevant literature show that the minimum or 25th percentile of SV is taken to be zero, and some of them even construct a dummy variable for whether or not the research is conducted, which shows that their samples include firms that have not been subject to institutional investors' SVs (Wang et al., 2020; Broadstock and Chen, 2021; Chen et al. 2021; Chen et al. 2021; Qi et al. 2021; Su et al. 2021; Chen et al. 2022; Jiang and Bai, 2022; Lai et al. 2022; Ling et al. 2022; Yang and Ma, 2022).

Since the end of 2012, the Shenzhen Stock Exchange (SZSE) has made it mandatory for listed companies to disclose detailed information about their institutional investors’ SVs. In the revised manuscript, we have added Table 1 Sample selection (Pages 9-10 and 23) that summarizes the sample selection procedure in section 4.1. The research sample includes Chinese A-share companies listed on the SZSE from 2013 to 2018. The study excludes financial companies, ST companies, and companies with missing key variables. The final sample includes 9,626 firm-year observations of 2,079 unique firms. Besides, drawing on Wang et al. (2022), we plot Figure 1 (Pages 9 and 35) in section 4.1 to provide a detailed picture of the distribution of firms that received institutional investors' SVs during the sample period.

The referenced studies above are as follows:

Jiang, X., and Q. Yuan. 2018. Institutional investors' corporate site visits and corporate innovation. Journal of Corporate Finance 48:148-168.

Wang, J. Y., G. Q. Liu, and Q. S. Xiong. 2020. Institutional investors' information seeking and stock price crash risk: nonlinear relationship based on management's opportunistic behaviour. Accounting and Finance 60 (5):4621-4649.

Broadstock, D., and X. Chen. 2021. Corporate site visits, private monitoring and fraud: Evidence from China. Finance Research Letters 40:101780.

Chen, X. Y., P. Wan, and M. S. Sial. 2021. Institutional investors' site visits and corporate social responsibility: Implications for the extractive industries. Extractive Industries and Society- An International Journal 8 (1):374-382.

Qi, Z., Y. Zhou, and J. Chen. 2021. Corporate site visits and earnings management. Journal of Accounting and Public Policy 40 (4):106823.

Su, F., X. Feng, and S. Tang. 2021. Do site visits mitigate corporate fraudulence? Evidence from China. International Review of Financial Analysis 78:101940.

Chen, X. Q., C. S. A. Cheng, J. Xie, and H. Y. Yang. 2022. Private communication and management forecasts: Evidence from corporate site visits. Corporate Governance-An International Review 30 (4):482-497.

Jiang, L., and Y. Bai. 2022. Strategic or substantive innovation? -The impact of institutional investors' site visits on green innovation evidence from China. Technology in Society 68:101904.

Lai, S., X. Li, S. Liu, and Q. S. Wang. 2022. Institutional investors’ site visits and corporate employment decision-making. Journal of Contemporary Accounting & Economics 18 (3):100332.

Ling, X., S. Yan, and L. T. W. Cheng. 2022. Investor relations under short-selling pressure: Evidence from strategic signaling by company site visits. Journal of Business Finance & Accounting 49 (7-8):1145-1174.

Yang, X., and Z. Ma. 2022. Institutional investors' corporate site visits and dividend payouts. International Review of Economics & Finance 80:697-716.

Wang, Q., S. J. Lai, X. P. Cao, and S. A. Liu. 2022. The effect of institutional investors' site visits: evidence on corporate cash holdings. Applied Economics 54 (41):4767-4781.

6. There must be a table explaining the sample history (initial, missing, deleted, out of scope firms etc.)

Response: 

Thank you for your valuable suggestions for the sample. In the revised manuscript, we have added Table 1 Sample selection (Pages 9-10 and 23) that summarizes the sample selection procedure in section 4.1. The research sample includes Chinese A-share companies listed on the SZSE from 2013 to 2018. The study excludes financial companies, ST companies, and companies with missing key variables. The final sample includes 9,626 firm-year observations of 2,079 unique firms. 

Please refer to the section 4.1 (Pages 9-10) and Table 1 (Page 23) for details.

7. More explanation is necessary about the theoretical background, how SV may or may not effect ICFS.

Response: 

Thank you very much for your constructive suggestions for the theoretical analysis. In the revised manuscript, we have reorganized the theory and hypothesis in section 3 (Pages 7-9), in which we strengthen the theoretical explanation of how institutional investors' SV affect ICFS. 

Please refer to the section 3 (Pages 7-9) for details.

8. The limitations of the study must be added.

Response: 

Thank you for your suggestions for the manuscript. In the revised manuscript, we have supplemented the limitations in section 6 Conclusions and policy implications. Please refer to the section 6 (Pages 18-19) for details.

Thank you once again for your meticulous review and constructive suggestions. 

Yours Sincerely, 

The authors

---

## [Decision Letter · Decision Letter 1]

27 Feb 2024

Institutional Investors' Site Visits and Investment-Cash Flow Sensitivity: Mitigating Financing Constraints or Inhibiting Agency Conflicts？

PONE-D-23-22387R1

Dear Dr. Zhan,

We’re pleased to inform you that your manuscript has been judged scientifically suitable for publication and will be formally accepted for publication once it meets all outstanding technical requirements.

Kind regards,

Jasman Tuyon, Ph.D., MBA

Academic Editor

PLOS ONE

Additional Editor Comments (optional):

Accepted as per reviewer recommendations

Reviewers' comments:

Reviewer's Responses to Questions

**Comments to the Author**

1. If the authors have adequately addressed your comments raised in a previous round of review and you feel that this manuscript is now acceptable for publication, you may indicate that here to bypass the “Comments to the Author” section, enter your conflict of interest statement in the “Confidential to Editor” section, and submit your "Accept" recommendation.

Reviewer #1: All comments have been addressed

Reviewer #2: All comments have been addressed

2. Is the manuscript technically sound, and do the data support the conclusions?

Reviewer #1: Yes

Reviewer #2: Yes

3. Has the statistical analysis been performed appropriately and rigorously? 

Reviewer #1: Yes

Reviewer #2: Yes

4. Have the authors made all data underlying the findings in their manuscript fully available?

Reviewer #1: Yes

Reviewer #2: Yes

5. Is the manuscript presented in an intelligible fashion and written in standard English?

Reviewer #1: Yes

Reviewer #2: Yes

6. Review Comments to the Author

Reviewer #1: (No Response)

Reviewer #2: The authors responded properly to the comments, there is no need for further development. Only journal format requirements should be considered at final stage.

7. PLOS authors have the option to publish the peer review history of their article (what does this mean?). If published, this will include your full peer review and any attached files.

Reviewer #1: No

Reviewer #2: No

---

## [Editor Report · Acceptance letter]

19 Mar 2024

PONE-D-23-22387R1 

PLOS ONE

Dear Dr. Zhan, 

I'm pleased to inform you that your manuscript has been deemed suitable for publication in PLOS ONE. Congratulations! Your manuscript is now being handed over to our production team.

Kind regards, 

on behalf of

Dr. Jasman Tuyon 

Academic Editor

PLOS ONE